# OpenAVE: Moving towards Open Set Audio-Visual Event Localization

Submission Id: 323*

## ABSTRACT

Audio-Visual Event (AVE) Localization aims to identify and classify video segments that are both audible and visible, a field that has seen substantial progress in recent years. Existing methods operate under a closed-set assumption and struggle to recognize unknown events in open-world scenarios. To better adapt to real-life applications, we introduce the Open Set Audio-Visual Event Localization task and propose a novel and effective network called OpenAVE based on evidential deep learning. To the best of our knowledge, this is the first effort to address this challenge. Our approach encompasses deep evidential AVE classification and event-relevant prediction, targeting the nuanced demands of open-set environments. Our approach includes deep evidential AVE classification and event-relevant prediction. The deep evidential AVE classification manages event classification uncertainty by extracting class evidence from segment-specific representations enriched with multi-scale context. To effectively distinguish between unknown events and background segments, event-relevant prediction utilizes positive-unlabeled learning. Futhermore, a learnable Gaussian-prior prediction branch is adopted to enhance the performance of event-relevant prediction. Experimental results demonstrate that OpenAVE significantly outperforms state-of-the-art models on the Audio-Visual Event dataset, confirming the effectiveness of our proposed method.

## CCS CONCEPTS

• **Computing methodologies → Hierarchical representations**; *Artificial intelligence*; *Computer vision*; Computer vision representations.

## KEYWORDS

Audio-visual event localization, Open set recognition, Cross-modality representation, Evidential deep learning

## 1 INTRODUCTION

Audio-Visual Event (AVE) Localization task [32] is one of hot topics in the fields of visual-audio scene understanding, which aims at simultaneously determining the presence of an event that is both audible and visible in a video segment from the arbitrarily untrimmed video, and classifying it into a certain event category.

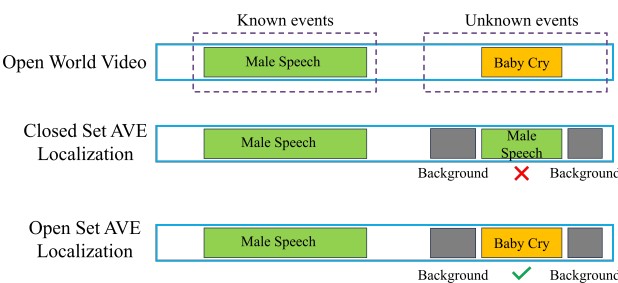

**Figure 1: Illustration of closed set AVE and open set AVE. Different from closed set AVE task, open set AVE could not only correctly recognize known events (*e.g.* Male Speech) in an untrimmed video but effectively reject the positive foreground event segment (*e.g.* Baby Cry) as the unknown in an open-world.**

It has attracted significant attention from the research community owing to its extensive application potentials, such as video summarization [42], action recognition [14], egocentric object detection [19], indoor navigation [6] and so on. In recent years, a plethora of audio-visual event localization approaches [22, 26, 36, 37, 44] have been proposed and have demonstrated remarkable performance. Despite their success, previous work mainly handle AVE task under the closed set assumption, which classify each event segment into one of the classes encountered during training. This closed set condition limits their application in real-world scenarios, since an input video whose classes are beyond the range of the training set will be misclassified as one of the known categories. Therefore, to tackle this problem, we consider a more challenging and practical AVE setting in this work, termed as **Open Set Audio-Visual Event Localization (OSAVE)**.

OSAVE aims to not only recognize known audio-visual events in each video segment but also to reject the unknown ones. As illustrated in Figure 1, given an untrimmed testing video containing a unknown event (*e.g.*, Baby Cry), traditional AVE methods fails to identify this unknown audio-visual event segment and tends to assign its label to a known class (*e.g.*, Male Speech). Therefore, categories of both known and unknown video segments are expected to be predicted in OSAVE task. Compared with traditional AVE problem, OSAVE is more challenging in two aspects: (1) The temporal nature of videos might result in the diversity and complexity of audio-visual events. Thus, an OSAVE model is required to learn more discriminative event representations of closed set categories but also be aware of what it does not belongs to a known class when the unknown event appears in an open-world scenario. (2) Previous AVE works [37, 38] could rely on segment-level label to correctly predict foreground events and the background. However, due to the lack of unknown event annotations in the open set setting, these

existing models fail to effectively differentiate unknown events and pure background from the mixture video segments in the inference stage.

In this paper, we propose a novel and effective framework (OpenAVE) for open set audio-visual event localization task. To tackle the first challenge, we design a multi-scale context perception module in the deep evidential AVE classification network to obtain more discriminative segment-specific features by taking advantage of richer temporal context, which is not only greatly help for known classification but also provides abundant and distinct information to estimate category uncertainty. To enable the model to know the unknown in the OSVAE task, we formulates it as an uncertainty estimation problem by leveraging evidential deep learning (EDL). EDL could directly use deep neural networks to predict a Dirichlet distribution of event class probabilities, which is informative to quantify the predictive uncertainty of audio-visual events so that the model could discover those high-uncertainty unknown ones.

To address the second challenge, we introduce an event-relevant prediction network designed to distinguish between unknown foreground events and background segments. Since the unknown foreground events without annotations are mixed with background segments, learning from labeled known audio-visual events and the mixture can be considered as a semi-supervised OSR problem. Therefore, we apply the positive-unlabeled learning (PU learning) algorithm by training a binary foreground-background classifier to discover potential foreground segments (known and unknown events) in the testing video. Besides, a learnable Gaussian-prior event-relevant branch leveraging local context is proposed to improve the smoothness of learning-based event-relevant scores, thus providing more reliable positive and negative samples for PU learning. Benefiting from evidential deep learning theory and PU learning, our proposed OpenAVE is not only practically flexible to implement but also more effective in distinguishing between known and unknown events. Based on the existing audio-visual event localization dataset (AVE dataset), we construct a new benchmark to evaluate our model and all baselines for OSAVE task. Extensive experiments show that our method outperforms state-of-the-art methods in realistic open-world scenarios. In summary, our main contributions are as follows:

- We propose an Open Set Audio-Visual Event localization network to identify unknown events within a video. To the best of our knowledge, this is the first work to address the Open Set Audio-Visual Event Localization task (OSAVE), a highly challenging yet significant task for open-world scenarios.

- To address the unique challenges of OSAVE, we propose a deep evidential AVE classification network to handle classification uncertainty effectively. This network leverages event class evidence from segment-specific features with rich temporal contexts. Additionally, an event-relevant prediction network is developed to facilitate the Positive and Unlabeled learning (PU learning), distinguishing unknown events from background segments efficiently.

- We conduct extensive experiments on a popular AVE benchmark, Audio-Visual Event dataset (AVE dataset), and compare our approach against various baselines. The experimental results clearly demonstrate that our proposed method

substantially outperforms these baselines, indicating a significant advancement in the field.

## 2 RELATED WORK

### 2.1 Audio-Visual Event Localization

**Audio-visual event (AVE) localization** aims to match audible and visible segments in untrimmed videos for identifying the simultaneous event of interest by relying on segment-level annotations during training. Tian et al. [32] first proposed an audio-visual event localization task and designed a dual multi-modal residual network to aggregate information over the auditory and visual modalities to handle cross-modality localization. Afterwards, to avoid the content of the two modalities being misaligned temporally, Wu et al. [35] proposed a dual attention matching module, which could better align visual and acoustic features of each segment while also capturing local temporal cues via the global cross-check mechanism. Xuan et al. [38] developed a novel cross-modal interaction framework comprising spatial, sequential, and cross-modal adaptive attention modules to comprehensively capture most event-related information. Xu et al. [37] introduced the relation-aware network that relied on cross-modality relation attention to establish useful intra-modality and inter-modality relationships. Besides, the positive sample propagation (PSP) method was proposed by Zhou et al. [44] to discover strong relevant audio-visual pairs and reduce background noise from weak related or negative pairs. Yan et al. [36] employed a temporal cross-modal background suppression scheme for the AVE task, effectively mitigating asynchronous audio-visual background noise while encouraging the model to learn closely related cross-modal information in the temporal dimension. Despite numerous explorations in audio-visual event localization, all these works were developed under the closed-set assumption. This assumption implies that testing videos only include a predefined set of known events. However, in the dynamically changing real world, where unknown classes are bound to emerge in untrimmed videos, this assumption becomes invalid. To overcome the constraints of the closed-set condition, we have specifically defined a novel task called open set audio-visual event localization (OSAVE). Additionally, we have devised a new framework for OSAVE based on evidential deep learning (EDL) and positive-unlabeled learning (PU learning). This framework is designed to address the more complex challenges associated with open-set AVE.

### 2.2 Open Set Recognition

**Open set recognition (OSR)** describes such a scenario where unknown classes not present in the training data appear in the inference stage. It requires the classifiers to accurately classify known categories in their training set and effectively reject samples that do not belong to any of the known ones. Scheirer et al. [27] first formalized the OSR problem and introduced a novel "1-vs-set machine" algorithm based on a binary SVM to identify unknown classes. With the tremendous progress of deep learning in the field of computer vision, Bendale et al. [3] proposed the first deep learning OSR approach, OpenMax, which predicts an unknown class by adapting statistical extreme value theory (EVT) to the $K$-class softmax classifier in the network. Recently, generative methods have also been

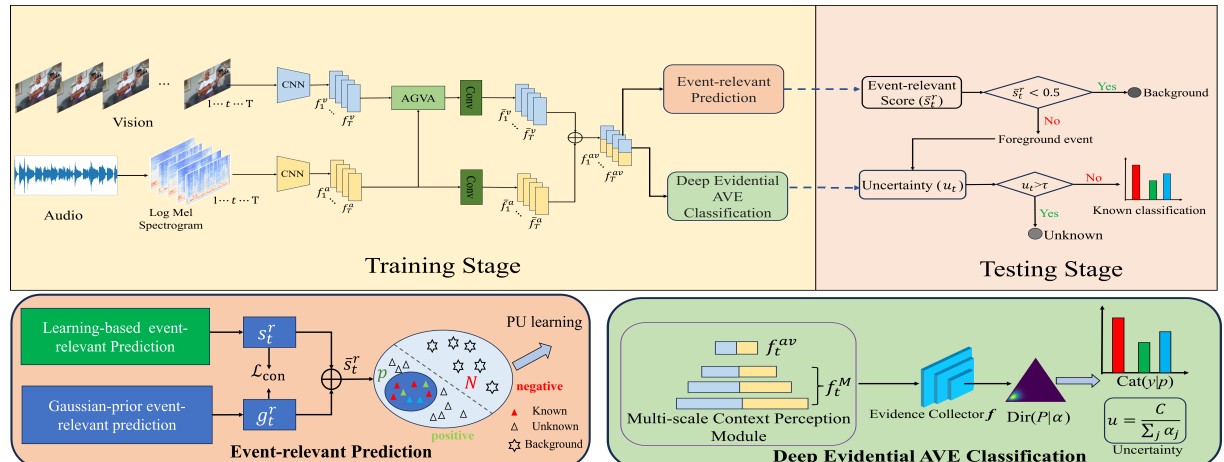

**Figure 2: The architecture of the proposed OpenAVE for Open Set Audio-Visual Event Localization (OSAVE) task. It contains audio and visual feature extraction, event-relevant prediction network and deep evidential AVE classification network. The deep evidential AVE classification network based on EDL and multi-scale context is used for the known/unknown judgment. The event-relevant prediction network mainly applies the positive-unlabeled learning (PU learning) for distinguish unlabeled unknown events from the background. In the inference stage, we leverage the uncertainty and event-relevant scores, video segments from the known and unknown classes, as well as background frames can be effectively distinguished in the OSAVE setting.**

explored to handle the OSR problem. For example, Neal et al. [24] introduced a counterfactual image generation method that leverages generated data close to training samples but not belonging to known classes to train an open set classifier. Other generative model-based works [25, 30, 39] borrowed the idea of feature reconstruction by utilizing the reconstruction error associated with the generator as an open set indicator to reject unknown ones. Prototype-based learning approaches [7, 9] aim to identify the unknown by calculating the maximum distance between the input example and the learned known class prototypes. Furthermore, probabilistic and evidential deep learning methods [11, 13, 23, 34] that estimate prediction uncertainty have emerged as potential approaches for improving OSR performance. Additionally, some works have extended the research field of evidential deep learning (EDL) from open set image recognition (OSR) to open set action recognition [1, 5, 40, 43], open set temporal action localization [2, 10, 18] and open set object detection/segmentation [12, 20, 21], yielding promising results. Unlike image samples, audio-visual events entail cross-modal associations and temporal dynamics. Therefore, our proposed Event-relevant prediction mechanism enables the distinction between unknown events and background segments from open-set setting.

## 3 THE PROPOSED METHOD

### 3.1 Notations and Preliminaries

**Problem formulation of OSAVE.** The Open Set Audio-Visual Event (OSAVE) localization task aims to predict which temporal segment of an input untrimmed video contains an audio-visual event and to determine the known category to which the event belongs, while also identifying and rejecting segments from novel classes as unknown. Formally, we divide a given video sequences

$\mathcal{V}$ into $T$ non-overlapping audio and visual pairs $\mathcal{V} = (A_t, V_t)_{t=1}^{T}$, where each segment is one second long. Here, $V_t$ represents the visual content, $A_t$ represents its synchronized audio counterpart for the $t$-th segment. The ground-truth label for each video segment is denoted as $y_t^c = \{y_t^c | y_t^c \in \{0, 1\}, \sum_{c=1}^{C+1} y_t^c = 1\} \in \mathbb{R}^{C+1}$, where $y_t^c \in \{0, 1\}$ indicates whether an event of category $c$ is present in the $t$-th segment. $C$ is the total number of known event categories, and $C + 1$ represents the background. $y_t^c = 1$ indicates the presence of an event of category $c$ in the $t$-th segment, while $y_t^c = 0$ represents its absence. The label for the entire video can be denoted as $Y = \{y_1^c, y_2^c, \cdots, y_T^c\} \in \mathbb{R}^{T \times C+1}$. It is worthy noting that the model only has access to the video data and the annotations of known events during training while the annotations of unknown events are not provided. During the inference phase, the learned model is required to predict a set of the event labels $\{\hat{y}_t^{\bar{c}}\}_{t=1}^N$, where $N$ is the number of the input video segments in $\hat{\mathcal{V}}$. Here, $\bar{c}$ represents an event category, where $\bar{c} \in \{1, 2, \cdots, C+1, U\}$, and $U$ denotes the unknown event class. In summary, this task involves labeling video data with audio-visual events, categorizing known events, and distinguishing them from unknown events based on the trained model's predictions.

**Audio and visual representations.** Following previous works [32, 37, 38], we also utilize pre-trained CNN models to extract audio-visual features $\{f_t^a, f_t^v\}$ for each segment $(A_t, V_t)$ in the video $\mathcal{V}$, where $t \in \{1, 2, 3, \cdots, T\}$. For the audio sequence input, we exploit the VGG-like network pre-trained on AudioSet to extract audio features $\{f_t^a\}_{t=1}^T \in \mathbb{R}^{T \times d_a}$, where $d_a$ denotes the dimension of audio feature vectors. Similarly, the visual representations $\{f_t^v\}_{t=1}^T \in \mathbb{R}^{T \times d_v \times (H*W)}$ are extracted by the ResNet-151 or VGG-19 model pre-trained on ImageNet, where $d_v$ denotes the visual feature dimension, and $H$ and $W$ are the height and width of the feature map. To reduce the background noise, we apply AGVA

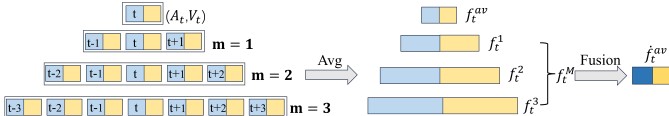

**Figure 3: The structure of the multi-scale context perception module (MSCPM).**

[32] to find visual regions that are relevant to the audio signals in each video segment. Then, the visual and audio features after AGVA are further input to two convolutional layers with ReLu to project the features from two modalities into the same embedding space. The processed audio and visual features are represented as $\{\bar{f}_t^a\}_{t=1}^T \in \mathbb{R}^{T \times D}$ and $\{\bar{f}_t^v\}_{t=1}^T \in \mathbb{R}^{T \times D}$, respectively. Finally, the average features of $\bar{f}_t^a$ and $\bar{f}_t^v$ are used as the joint multi-modality representation $f_t^{av} \in \mathbb{R}^{1 \times D}$ for labeling the video segment.

## 3.2 The Architecture

The architecture of our approach primarily consists of two main components: a deep evidential AVE classification network and an event-relevant prediction network. Given an untrimmed video, the deep evidential AVE classification network is tasked with identifying and classifying audio-visual events by leveraging evidential deep learning to manage uncertainty and enhance reliability. This network processes both audio and visual streams to extract features that are combined to form a comprehensive multi-modal understanding of each event segment. Concurrently, the event-relevant prediction network focuses on determining the relevance of each video segment to the detected events. This network employs positive-unlabeled learning to distinguish between segments that contain relevant events and those that do not, effectively filtering out irrelevant or background segments. By integrating these two networks, our approach not only accurately classifies known audio-visual events but also adeptly identifies segments containing new or previously unrecognized event types, enhancing the model's applicability in dynamic, real-world environments where unseen events may occur.

As depicted in Figure 2, we first extract audio and visual features $\{\bar{f}_t^a\}_{t=1}^T$ and $\{\bar{f}_t^v\}_{t=1}^T$, respectively, and then integrate them to generate multi-modality presentations $\{f_{av}\}_{t=1}^T$. Next, the multi-modality features $\{f_t^{av}\}_{t=1}^T$ are sent to deep evidential AVE classification network and event-relevant prediction network respectively to jointly determine known and unknown events in the inference stage. In the deep evidential AVE classification network, we first apply a multi-scale context perception module to enable individual segment to integrate rich temporal contextual information from neighborhood video segments. Then the evidence collector $f$ on top of the multi-scale context perception module is used to predict class-wise evidence, which formulates a Dirichlet distribution so that event classification probabilities $p$ and the predictive uncertainty $u$ of each video segment can be determined. To distinguish unknown foreground events from the background, the event-relevant prediction network is learned by the positive-unlabeled learning (PU learning). To further enhance the event-relevant scores $\bar{s}_t^r$ (predicting the foreground event and background) to construct positive

and negative sets in PU learning, we add the Gaussian-prior event-relevant prediction branch to improve the smoothness of predicted video segments. Moreover, a consistency loss is utilized to reduce the disagreement between the learning-based event-relevant scores $s_t^r$ and Gaussian-prior event-relevant scores $g_t^r$. We now detail the specifics of our model.

## 3.3 Deep Evidential AVE Classification

### 3.3.1 Multi-scale Context Perception.
An untrimmed video might involve several known or unknown events usually ranging across multiple temporal scales in the open-world scenarios. Although some previous AVE approaches [15, 29, 32, 37, 44] adopt Bi-LSTMs [41] or Transformer [33] to establish temporal relationship in the uni-modal or cross-modal for localizing short as well as long-scale events, these model severely ignore the importance of local contextual perception. In fact, neighboring video segments of an event commonly contain more critical contextual cues related to it compared to distant video segments. Moreover, due to an audio-visual event may range diverse duration, sensing temporal contexts in diverse ranges is very essential. Therefore, we argue that endowing individual video segment with the ability of perceiving multi-scale local temporal contexts from neighborhood video segment is very crucial for AVE task.

To this end, a multi-scale context perception module (MSCPM) is proposed to enhance each individual event representation by making use of multi-scale temporal contextual cues, which is illustrated in Figure 3. Specifically, we assign each video segment $(A_t, V_t)$ with a series of $t$-centered multi-scale video segments that provide different temporal context-aware features to extend segment-specific perception ranges. Each $t$-centered multi-scale audio-visual segments are denoted as temporal sequence set $Z_t = \{(s_t^m, e_t^m)|m \in [1, M])\}$, where $M$ is the maximum of temporal extension range, $s_t^m = \max(0, t - m)$ and $e_t^m = \min(t + m, T)$ represents the starting and ending points of the temporal boundary, respectively. After, we extract the average features $f_t^m$ of every temporal sequence $(s_t^m, e_t^m) \in Z_t$, which is defined as:

$$f_t^m = \frac{1}{e_t^m - s_t^m} \sum_{s_t^m \le i \le e_t^m} f_{t=i}^{av} \tag{1}$$

To extend the perception range of the audio-visual segment $(A_t, V_t)$, we apply feature enhancement to integrate its audio-visual representation and each temporal context $f_t^m$. In general, each individual event gains useful contextual information mainly from extended video segments that are highly relevant to it. To effectively implement feature enhancement and reduce the interference of irrelevant context, this study proposes a weighted feature fusion mechanism. To be specific, we first measure the similarity between the segment features $f_t^{av}$ and each extended video sequence features $\{f_t^m\}_{m=1}^M$, and achieve the cosine similarities $\omega_t^m$, which can be given by:

$$\omega_t^m = \cos(\frac{f_t^{av}}{|f_t^{av}|}, \frac{f_t^m}{|f_t^m|}) \tag{2}$$

Then the $f_t^{av}$ and average feature representations of each weighted extended video segments are fused, and we obtain new audio-visual features $\dot{f}_t^{av}$ with extended-range perception, which is formulated

as:

$$\dot{f}_t^{av} = (1 - \gamma_t)\psi_1(f_t^{av}) + \gamma_t \sum_{m \in M} \delta(\omega_t^m)\psi_2(f_t^m) \tag{3}$$

Where $\psi_1$ and $\psi_2$ are two fully-connected layers for feature embedding, respectively; $\delta$ represents the Softmax operation with a temperature factor. $\gamma_t \in [0, 1]$ is the fusion weight representing the scaled average of the cosine similarities $\{\omega_t^m\}_{m=1}^M$, which can be calculated by:

$$\gamma_t = \frac{1}{2|M|} \sum_{m \in M} (\omega_t^m + 1) \tag{4}$$

*3.3.2 Uncertainty-aware AVE Classification.* In contrast to traditional closed-set audio-visual event localization, our model requires estimation of classification uncertainty to determine unknown events in the video. We mainly employ evidential deep learning (EDL) to learn an open set event classifier with quantified classification uncertainty. To achieve this, we build a Dirichlet distribution $Dir(p|\alpha)$ over class probabilities $p \in \mathbb{R}^C$, where $\alpha$ is the Dirichlet strength. The goal of the EDL is to predict $\alpha$ by deep neural networks (DNN). Our model is optimized by minimizing the negative log-likelihood of each audio-visual pair $\{(A_t, V_t), y_t^c\}_{t=1}^T$ in the video, which can be given by:

$$\mathcal{L}_{EDL} = \sum_j^C y_c^{(t)}(\log(S^{(t)}) - \log(\alpha_j^{(t)})) \tag{5}$$

Where $y_c^{(t)}$ is an one-hot $C$-dimensional label for a video segment $(A_t, V_t)$ and $S^{(t)} = \sum_j^C \alpha_j$ is the total strength over $C$ event categories. Based on Subjective Logic theory (SL) and the evidence theory, $\alpha_j$ is linked to the learned evidence $e_j$ by the equality $\alpha_j = e_j+1$, where $e_j \in \mathbb{R}_+^C$ can be represented as $e_j = g(f(\dot{f}_t^{av}; \theta))$. Here, $f$ can be seen as an evidence collector, which represents the output of a deep neural network (DNN) parameterized by $\theta$. $g$ is the evidence function, *e.g.*, Exp, Softplus or ReLU, to keep the collected evidence $e_j$ non-negative. In the testing, the expected classification probability of each event category is $\mathbb{E}[p_j] = \alpha_j/S$ and the classification uncertainty $u$ is estimated by $u_j = C/S$. $\mathbb{E}[\cdot]$ is to take the mean loss values over the input samples. Note that the uncertainty $u$ is inversely proportional to the total evidence of all known categories, therefore it reflects the probability that a video segment belongs to the unknown event.

## 3.4 Event-relevant Prediction

When a given video contains unknown events, the mixture of unknown foreground segments and pure background makes it challenging for the model to distinguish between them solely through uncertainty-aware classification. Therefore, predicting event-relevant scores that indicates the likelihood of a video segment is the foreground is crucial. We also notice the fact that samples from known categories are positive while the mixture of 'background' includes 'positive-unlabeled' data. This intrinsically is a semi-supervised learning problem referred to as **positive and unlabelled learning** (PU learning) which relies on learned knowledge from positive data to relabel unknown samples. To accurately identify the positive unknown samples from the 'background', a simple but very effective PU learning method is leveraged in this paper.

We utilize $s_t^r \in [0, 1]$ generating from the learning-based event-relevant prediction to represent the predicted event-relevant score of each audio-visual pair $(A_t, V_t)$. Note that the learning-based event-relevant prediction branch utilizes a simple binary classifier, consisting of a 2-dimensional fully-connected layer with the sigmoid function, to differentiate between foreground events and the background. A binary cross-entropy (BCE) loss is exploited as the training loss of event-relevant prediction network, which can be denoted as:

$$\mathcal{L}_r = -\frac{1}{|\mathcal{P}|} \sum_{s_t^r \in \mathcal{P}} \log s_t^r - \frac{1}{|\mathcal{N}|} \sum_{s_t^r \in \mathcal{N}} \log(1 - s_t^r) \tag{6}$$

where $\mathcal{P}$ and $\mathcal{N}$ are the positive and negative sets in a training batch. The positive set $\mathcal{P} = \{s_t^r | y_t^c \leq C\}$ directly consists of the data belong to known classes while the negative samples are difficult to determine owing to unknown events appearing in unlabeled data. To tackle this problem, we denote those samples $\{s_t^r | y_t^c = C + 1\}$ as the unlabeled background set $\mathcal{U}$, and then sort the $\mathcal{U}$ in ascending order. Finally, the top-$k$ examples from the $\mathcal{U}$ are selected to construct the most likely negative set $\mathcal{N}$. This simple loss of event-relevant prediction network will push the probably pure background segments far away from positive audio-visual event pairs.

Since the learning-based event-relevant score $s_t^r$ is typically determined on a segment-level basis without leveraging contextual information from neighboring video segments, we are considering utilizing local context to enhance it. This approach aims to achieve more reliable event-relevant prediction and further improve foreground discrimination. The primary motivation behind using local context is that the predicted event-relevant results should keep a locally consistency. To implement this prior, we introduce the learnable Gaussian masks to generate event-relevant scores. We mainly add a additional Gaussian-prior event-relevant prediction branch in the event-relevant prediction network, which predicts Gaussian kernels $(\sigma_t, \mu_t)_{t=1}^T$ to model event-relevant scores for each segment in the video. We obtain a set of segment-specific local Gaussian masks $G = \{G_j\}_{t=1}^T$ by the predicted Gaussian parameters $(\sigma_t, \mu_t)_{t=1}^T$. The Gaussian-prior event-relevant score $g_t^r$ is generated by choosing the values from the $t$-th Gaussian mask $G_t$, which is formalized as:

$$g_t^r = G_t = \exp(-\frac{\beta(j/T - \mu_t)^2}{\sigma_t^2})_{j=1}^T \tag{7}$$

Where $\beta$ represents the variance of the segment-specific local Gaussian masks $G$. Although Gaussian-prior event-relevant scores $\{g_t^r\}_{t=1}^T$ can provide locally smooth results for predicted audio-visual pairs, directly integrating them with the learning-based event-relevant scores $\{s_t^r\}_{t=1}^T$ will make the model worse. We assume that this issue occurs due to the disagreement between these two event-relevant scores on the same time step. To address this problem, an event-relevant consistency loss $\mathcal{L}_{con}$ is proposed, which is defined as:

$$\mathcal{L}_{con} = \sum_t (s_t^r - g_t^r)^2 \tag{8}$$

Under the help of the consistency loss, the learnable Gaussian-prior event-relevant scores could be effectively complemented to learning-based event-relevant scores to improve the smoothness

Table 1: Experimental results on Audio-Visual Event dataset (AVE dataset) evaluated by the AUPOC, AUPR and FAR@95. The closed set classification accuracy (ACC) is also presented for reference. All approaches are trained on three splits of the AVE training set, and evaluated on the entire AVE test set containing known and unknown classes. Best results are shown in bold.

| Methods | Audio-Visual Event | | | | | | | | | | | |
| | Split I | | | | Split II | | | | Split III | | | |
| | AUROC | AUPR | FAR@95(↓) | ACC | AUROC | AUPR | FAR@95(↓) | ACC | AUROC | AUPR | FAR@95(↓) | ACC |
| SoftMax(VGG-19) | 54.46 | 27.86 | 87.12 | 46.77 | 51.20 | 27.53 | 89.27 | 49.34 | 53.13 | 31.45 | 88.78 | 49.78 |
| OpenMax(VGG-19) | 54.21 | 33.64 | 85.24 | 42.52 | 48.36 | 30.28 | 89.91 | 46.52 | 51.33 | 32.44 | 88.29 | 47.28 |
| RPL(VGG-19) | 67.26 | 46.14 | 79.31 | 55.49 | 71.34 | 51.74 | 76.29 | 53.42 | 68.25 | 49.48 | 78.32 | 57.56 |
| ARPL(VGG-19) | 73.37 | 52.13 | 74.88 | 58.29 | 73.15 | 56.39 | 74.24 | 59.34 | 72.02 | 56.35 | 75.25 | 60.51 |
| Ours(VGG-19) | **76.68** | **57.34** | **71.39** | **70.40** | **79.89** | **67.61** | **55.82** | **68.87** | **85.62** | **70.24** | **54.46** | **71.40** |
| SoftMax(ResNet-151) | 59.76 | 32.90 | 81.32 | 53.17 | 56.70 | 32.87 | 84.23 | 56.53 | 56.24 | 32.21 | 84.78 | 50.30 |
| OpenMax(ResNet-151) | 56.62 | 36.43 | 83.34 | 45.46 | 54.56 | 34.89 | 86.72 | 48.20 | 54.60 | 35.24 | 86.26 | 52.56 |
| RPL(ResNet-151) | 70.46 | 49.54 | 75.21 | 58.48 | 73.54 | 54.68 | 72.59 | 55.29 | 71.89 | 51.56 | 71.54 | 60.01 |
| ARPL(ResNet-151) | 78.23 | 54.07 | 69.83 | 61.52 | 78.21 | 59.58 | 69.12 | 62.82 | 76.24 | 59.67 | 70.87 | 62.54 |
| Ours(ResNet-151) | **80.45** | **60.58** | **67.06** | **73.70** | **85.45** | **71.58** | **50.73** | **71.41** | **89.36** | **74.24** | **48.56** | **74.25** |

of predicted segments and provide more accurate foreground predictions. The average scores of $s_t^r$ and $g_t^r$ are fused to produce $\bar{s}_t^r$, which replaces original learning-based event-relevant score $s_t^r$ as the new event-relevant score. This new event-relevant score is used to construct positive and negative sets for PU learning. Finally, we optimize the **Event-relevant Prediction Network**, as shown in Figure 2, using the combined loss $\mathcal{L}_{ER} = \mathcal{L}_{con} + \mathcal{L}_r$.

## 3.5 Learning and Inference

**training.** By combining all the optimization objectives defined by Eqs.(5)(6)(8), the final weighted sum of multi-task training loss $\mathcal{L}_{total}$ is obtained as:

$$\mathcal{L}_{total} = \lambda \mathcal{L}_{EDL} + \mathcal{L}_{ER} \qquad (9)$$

Where $\lambda$ is a hyperparameter to balance training loss.

**Inference.** For a given untrimmed video, each audio-visual segment input is successively fed into the trained OpenAVE model, which generates the classification labels $\hat{y}_t = \arg\max_{j \in [1,2,\cdots,C]} \mathbb{E}[p_{tj}]$, the classification uncertainty $u_t$ and an event-relevant prediction score $\bar{s}_t^r$. Relying on the obtained $u_t$ and $\bar{s}_t^r$, a positive foreground audio-visual segment ($\bar{s}_t^r > 0.5$) can be accepted as the known class label $\hat{y}_t$ if the $u_t < \tau$, else it is rejected as the unknown, where $\tau$ represents a pre-defined outlier threshold. The entire inference procedure of our model is shown in Algorithm 1.

## 4 EXPERIMENTS

### 4.1 Experimental Setup

**Datasets.** We evaluate our method and all baseline models on a public audio-visual event localization benchmark dataset: the Audio-Visual Event (AVE) dataset. The AVE dataset is a subset of the AudioSet [16], which contains 4,143 videos covering 28 event categories, e.g., airplane flying, dog barking and church bell. Each video lasts for 10 seconds and contains at least one 2-second audio-visual event. The dataset is divided into 3,339 training videos, 402 validation videos, and 402 testing videos. To facilitate open set evaluation, we randomly select 3/4 of the event categories from the AVE training set as known data, while retaining all AVE validation and testing data containing both known and unknown categories.

---

**Algorithm 1** Inference Procedure

**Require:** Untrimmed test video $\hat{\mathcal{V}}$.
**Require:** The trained OpenAVE model.
**Require:** Threshold $\tau$ obtained from training data.
**Output:** Prediction Set $\hat{Y} = \{\hat{y}_1^{\bar{c}}, \hat{y}_2^{\bar{c}}, \hat{y}_t^{\bar{c}}, \cdots, \hat{y}_N^{\bar{c}}\}$ in the video $\hat{\mathcal{V}}$.
 1: Video data pre-processing.
 2: Predict the closed set prediction score $p_{tj}$, event-relevant scores $\bar{s}_t^r$ and classification uncertainty $u_t$ of each audio-visual pair $(A_t, V_t)_{t=1}^N$ in the video $\hat{\mathcal{V}}$.
 3: **while** $\{(A_t, V_t)\}_{t=1}^N \in \hat{\mathcal{V}}$ **do**
 4:    **if** $\bar{s}_t^r < 0.5$ **then**
        the video segment $(A_t, V_t)$ is a **background** and $\hat{y}_t^{\bar{c}} = C+1$; continue.
 5:    **end if**
 6:    **if** $u_t < \tau$ **then**
        the video segment $(A_t, V_t)$ is a **known** class by $\hat{y}_t^{\bar{c}} = \arg\max_j \mathbb{E}[p_{tj}], j \in [1, \cdots, C]$.
 7:    **else**
        the video segment $(A_t, V_t)$ is the **unknown** and $\hat{y}_t^{\bar{c}} = U$.
 8:    **end if**
 9: **end while**

---

This random selection process is repeated three times, resulting in three different open set data splits. Detailed dataset information is provided in our supplementary materials.

**Evaluation metrics.** In the experiment, the evaluation metrics are divided into closed set and open set metrics. For closed set evaluation metric, we follow previous works [32, 37] and utilize the overall classification accuracy (ACC) to assess closed set AVE performance. To adapt OSAVE evaluation, we introduce the Area Under the Receiver Operating Characteristic (AUROC) curve and the Area Under the Precision-Recall (AUPR) as open set evaluation metrics. These metrics evaluate the performance of detecting unknown events from known events. Additionally, we apply the False Alarm Rate at True Positive Rate of 95% (FAR@95) to address practical operational implications.

**Implementation details.** We use the ResNet-151 [17] and VGG-19 [28] pre-trained on ImageNet as the visual feature extractor, respectively. For each one-second video segment, We sample 16 frames and extract their visual features respectively. Then, we use the average feature maps of these frames to produce segment-level visual feature maps, resulting in the visual dimension of $7 \times 7 \times 2048$ or $7 \times 7 \times 512$ for each segment. For audio representations, we first process the raw audio into log-mel spectrograms. Then we utilize the VGG-like network pre-trained on AudioSet to extract 128-dimensional audio features. The projected feature dimension $D$ is 512, and the evidence function $g$ is the Exp. We empirically set the loss weight value $\lambda$ to 2 and the number of negative samples $k$ to 40. The maximum value of pre-defined extended length $M$ is 3. Similar to previous studies [1, 31], we determine the threshold $\tau$ of uncertainty $u_t$ in Algorithm 1 by ensuring that 95% training data is recognized as known. Our model is trained for 30 epochs on AVE dataset with a batch size of 16. We utilize the Adam optimizer with an initial learning rate of $1e-4$ and a weight decay $1e-5$ for model optimization. The overall experiments are conducted on an RTX 3090 GPU. In our ablation studies, we use the ResNet-151 as the visual feature extractor.

## 4.2 Comparisons with State-of-the-arts

To evaluate the performance of our proposed OpenAVE model, we compare it with the following baselines: (1) **SoftMax**: A standard confidence-based method for open set recognition that utilizes the softmax scores to identify the unknown. (2) **OpenMax**: This approach extends the softmax classification scores by appending unknown scores, as introduced by OpenMax [4], during testing. (3) **RPL** [9]: A novel OSR method that primarily employs Reciprocal Point Learning (RPL) to learn compact and discriminative representations for effectively identifying unknown classes. (4) **ARPL** [8]: Similar to RPL [9], this method uses Adversarial Reciprocal Points Learning (ARPL) to classify known and unknown classes. We separately train our method and all baselines on three different splits of the AVE training set and evaluate their performance on the testing dataset.

The experimental results on the AVE dataset are reported in Table 1. These results clearly demonstrate that our approach, OpenAVE, consistently and significantly outperforms all comparative models by large margins across both closed and open set AVE metrics. For instance, using the VGG-19 backbone, our method achieves the best AUROC score of 85.62% on AVE dataset split III, surpassing the state-of-the-art ARPL score of 72.02%. Although OpenMax and RPL are also considered recent state-of-the-art approaches for Open Set Recognition (OSR), it is evident that their performances lag considerably behind our method. Notably, the closed-set accuracy (ACC) of OpenMax is dramatically lower compared to other models. This is attributed to OpenMax's strategy of modifying the activation function before the softmax layer and introducing a novel unknown class, which could adversely affect the predictions of known events. When utilizing ResNet-151, our model shows even higher gains, improving AUROC scores by 7.24% on AVE dataset split II compared to ARPL. Despite variations in experimental outcomes across different splits of the AVE dataset, these comparisons robustly highlight the superior performance of OpenAVE in tackling the Open Set Audio-Visual Event Localization (OSAVE) task.

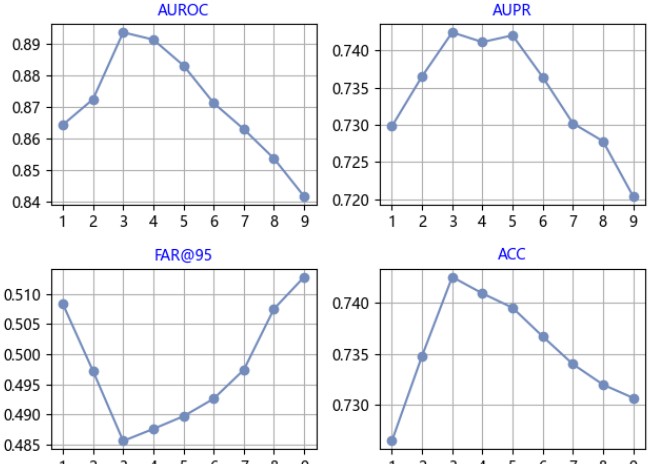

**Figure 4: Ablation study on the parameter $M$ in the multi-scale contextual perception module.**

**Table 2: Ablation study on the components of our proposed OpenAVE. MSCPM represents the multi-scale context perception module, ERP denotes the event-relevant prediction network that is learned by PU learning, and GP is a Gaussion-prior event-relevant prediction branch in the event-relevant prediction network. Best results are shown in bold.**

| | MSCPM | ERP | GP | AUROC | AUPR | FAR@95($\downarrow$) | ACC |
|---|---|---|---|---|---|---|---|
| 1 | $\times$ | $\checkmark$ | $\checkmark$ | 82.67 | 69.52 | 53.27 | 70.41 |
| 2 | $\checkmark$ | $\times$ | $\times$ | 75.93 | 57.53 | 72.31 | 61.67 |
| 3 | $\checkmark$ | $\checkmark$ | $\times$ | 86.58 | 71.26 | 50.21 | 72.03 |
| 4 | $\checkmark$ | $\checkmark$ | $\checkmark$ | **89.36** | **74.24** | **48.56** | **74.25** |

## 4.3 Ablation Study

*4.3.1 The effectiveness of Components.* In this section, we conducted ablation studies on the AVE test set to validate the effectiveness of different components in our OpenAVE, and the ablation results are presented in Table 2. It is evident that all proposed components significantly contribute to enhancing our model's performance. In particular, the multi-scale context perception module (MSCPM) exhibits the highest impact on the AUROC metric, leading to a notable performance improvement of 6.69%. Note that without MSCPM (row 1), we replace the fused multi-scale temporal features $\hat{f}_t^{av}$ by the individual segment-level audio-visual features $f_t^{av}$. The event-relevant prediction network (ERP) contributes the most to all metrics. This fully demonstrates ERP plays a very important role in our model, which mainly utilizes PU learning to effectively discover potential unknown foreground events to help the model well identify known class and unknown ones. Besides, the Gaussion-prior event-relevant prediction branch (GP) further bring the beneficial improvement on all metrics. In summary, these results sufficiently demonstrate that our proposed components are effective and indispensable for OSAVE task.

**Table 3: Ablation study on temporal baselines**

| Methods | AUROC | AUPR | FAR@95(↓) | ACC |
|---|---|---|---|---|
| Bi-LSTM | 83.75 | 71.12 | 56.17 | 71.47 |
| Transformer | 81.45 | 70.23 | 58.68 | 73.76 |
| Ours | **89.36** | **74.24** | **48.56** | **74.25** |

*4.3.2 The effectiveness of temporal contexts.* In this part, we conduct an ablation experiment to explore the importance of multi-scale temporal contextual perception. We primarily compare our proposed multi-scale contextual perception module (MSCPM) with Bi-LSTM [41] and Transformer [33] on the AVE testing set, and the experimental results are demonstrated in Table 3. It is worth mentioning that only the multi-scale context perception module was replaced by Bi-LSTM and Transformer which contains two encoder layers, while the rest of the components remained consistent throughout the experiment. It can be observed that our proposed MSCPM achieves the best performance by taking advantage of multi-scale temporal contexts. It outperforms both Bi-LSTM and Transformer by a significant margin across all metrics. For instance, our MSCPM achieves an AUROC score of 89.36%, markedly surpassing the Transformer's score of 81.45%. However, it is worth noting that open set evaluation metrics of the Transformer were lower than those of Bi-LSTM. This discrepancy arises from the Transformer's strong global temporal modeling capability, which could lead to overfitting to known categories. Despite the well-implemented Bi-LSTM and Transformer in previous closed-set AVE tasks, our experiment effectively illustrates how global temporal information fails to provide sufficiently discriminative segment-specific features to aid the uncertainty classifier in rejecting unknown events. Furthermore, we analyzed the impact of different temporal context scales on our model's performance, as depicted in Figure 4. The extension ranges, denoted by $M$, were varied from one to nine based on the number of video segments. Our model achieved optimal performance across all metrics when $M$ was set to 3. However, as the extension ranges increased, the model's performance on metrics began to decline. This decline can be attributed to the aggregation of more event-irrelevant contextual information into the current video segment as the temporal extension scale increases, thereby impeding the model's ability to extract valuable classification evidence to quantify event predictive uncertainty.

**Table 4: Ablation study on the consistency loss in the event-relevant prediction network**

| Method | AUROC |
|---|---|
| OpenAVE w/o consistency loss | 87.56 |
| OpenAVE w. consistency loss | **89.36** |

*4.3.3 The effectiveness of the consistency loss.* we further explore the effectiveness of the proposed consistency loss in the event-relevant prediction network. As shown in Table 4, we can see that the performance declines by 1.80% on the AUROC metric without the consistency loss. This further verifies our hypothesis that the two different kinds of event relevant prediction scores require to be aligned for the best performance.

*4.3.4 Visualization Analysis of Event-Relevant Prediction and Uncertainty Classification.* To demonstrate the quality of the learned event-relevant prediction and AVE uncertainty classification, we visualized their density distributions on the AVE test set in Figure 5. It is evident from Figure 5.(a) that foreground events have high event-relevant scores, while the background events are associated with low prediction scores. The dominant modes depicted in Figure 5.(b) indicate that audio-visual events belonging to known classes are assigned low uncertainty scores, whereas those of unknown classes exhibit high uncertainty. These observations align well with the expectations of our OpenAVE model.

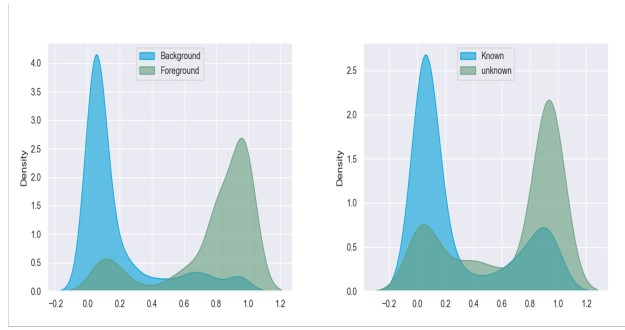

(a) Event-relevant Prediction      (b) Uncertainty Classification

**Figure 5: Distributions of event-relevant prediction and uncertainty classification. The two figures show significant separation between the foreground events and background segments by event-relevant prediction scores, as well as the separation between the known and unknown audio-visual events by uncertainty classification results.**

## 5 CONCLUSION

In this paper, we investigate the Open Set Audio-Visual Event Localization (OSAVE) task for the first time, which involves recognizing known events while simultaneously rejecting unknown audio-visual events in open-world scenarios. The OSAVE task presents greater challenges than the traditional AVE task due to the presence of both background segments and unknown foreground events, compounded by the inherent uncertainty of the events. To address this, we propose a novel network for open set audio-visual event localization, which comprises an event-relevant prediction network and a deep evidential AVE classification network. In the proposed OpenAVE, the deep evidential AVE classification network utilizes Evidential Deep Learning (EDL) to manage event classification uncertainty and employs a Multi-Scale Context Perception Module (MSCPM) to derive segment-specific representations that provide more discriminative evidence for the EDL classifier. The event-relevant prediction network learns to distinguish unknown events from the background using the Positive-Unlabeled learning algorithm (PU learning). Futhermore, a learnable Gaussian-prior branch is integrated into the event-relevant prediction network to enhance the reliability of positive and negative samples for PU learning. Extensive experiments on the Audio-Visual Event benchmark (AVE dataset) demonstrate that our approach achieves state-of-the-art performance in OSAVE.

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
