# OpenReview forum: "OpenAVE: Moving towards Open Set Audio-Visual Event Localization"
_acmmm.org/ACMMM/2024/Conference — MM2024 Oral_

### Official Review · Reviewer_KmcA · 2024-05-16

**Rating:** 6
**Confidence:** 2

**Summary:**

The paper introduces "OpenAVE," a novel framework for Open Set Audio-Visual Event (OSAVE) Localization, addressing the challenge of recognizing and classifying video segments containing concurrent audible and visual cues in real-world, open-world scenarios. This is a pioneering work in the field, as it shifts the focus from traditional closed-set assumptions to handling unknown events. The OpenAVE model incorporates two key components: a deep evidential AVE classification network and an event-relevant prediction network. The former utilizes evidential deep learning combined with multi-scale context to manage classification uncertainty for known events while also identifying unknown ones. The latter employs positive-unlabeled (PU) learning to distinguish unknown foreground events from background segments, further enhanced by a learnable Gaussian-prior branch for smoother learning-based event relevance scores. Comprehensive evaluations on the Audio-Visual Event (AVE) dataset demonstrate that OpenAVE surpasses state-of-the-art models, validating its efficacy for OSAVE.

**Strengths:**

Innovation: The paper presents the first dedicated solution for open set audio-visual event localization, filling a crucial gap in the literature where most methods operate under closed-set constraints.
Evidential Deep Learning Integration: By leveraging evidential theory, OpenAVE provides a robust way to quantify uncertainty in event classification, which is particularly valuable in the presence of unseen events.
Event-Relevant Prediction Network: The introduction of PU learning, alongside the Gaussian-prior branch, enhances the model's ability to discern unknown events from backgrounds, contributing to more accurate open-world event detection.
Comprehensive Evaluation: The authors establish a new benchmark using the AVE dataset for the OSAVE task, showcasing the superiority of OpenAVE through extensive experiments under realistic conditions.

**Limitations:**

There is repeated content in the abstract. “Our approach encompasses…prediction.” V.S. “Our approach includes…prediction.”
Generalizability Concerns: While the proposed model shows strong performance on the AVE dataset, the generalizability to other datasets or real-life scenarios with different event distributions and noise levels is not thoroughly explored.
Complexity and Scalability: The deep evidential and event-relevant prediction networks, while effective, may introduce computational complexity that could limit scalability for large-scale or real-time applications.
Unknown Event Understanding: Although the model distinguishes unknown events from known ones and background, it does not inherently provide insights into what these unknown events might be or how they can be characterized, limiting its utility for event discovery and taxonomy expansion.
Lack of Qualitative Analysis: While quantitative results are comprehensive, qualitative examples demonstrating how the model performs in challenging cases or the types of errors it makes would enrich the understanding of its capabilities.
Overall, OpenAVE represents a significant step forward in audio-visual event localization by tackling open-set recognition, offering a solid foundation for future advancements in understanding complex multimedia content in open-world settings. However, further research could address its limitations, focusing on enhancing scalability, exploring event characterization for unknowns, and validating performance across diverse datasets and real-life scenarios.

**Suitability:**

3

---

### Official Review · Reviewer_WBR5 · 2024-05-23

**Rating:** 6
**Confidence:** 3

**Summary:**

The proposed method significantly enhances performance in the field of Open Set Audio-Visual Event Localization (OSAVE) by effectively recognizing known events while rejecting unknown ones in open-world scenarios. The introduced model, OpenAVE, integrates an event-relevant prediction network and a deep evidential AVE classification network, effectively managing event classification uncertainty and distinguishing between known and unknown events.

**Strengths:**

This paper is well written.

**Limitations:**

In the “RELATED WORK” section, while the paper offers a comprehensive overview of existing methods in Open Set Recognition (OSR), it would benefit from providing more direct comparisons between different approaches, particularly in terms of their effectiveness in handling uncertainty and unknown classes. This would help readers better understand the strengths and limitations of each method and provide clearer justification for the proposed Event-relevant prediction mechanism.

In the "The Proposed Method" section, it would be beneficial to provide a more detailed explanation of why the Dirichlet distribution is employed and how it enhances the method's effectiveness. By elaborating on specific instances or scenarios where the Dirichlet distribution effectively models uncertainty in audio-visual event localization, readers can better understand its practical advantages and the rationale behind its utilization in the proposed approach.

In the “The Proposed Method” section, the inclusion of Gaussian-prior event-relevant scores enhances local context awareness. However, authors could further elucidate how these scores complement the learning-based event-relevant scores to improve overall predictions, providing clarity on the synergy between the two approaches.
In the "EXPERIMENTS" section, the study evaluates the effectiveness of components in the OpenAVE model. Results highlight MSCPM's key role, yielding a substantial AUROC improvement. ERP and GP also contribute significantly. MSCPM outperforms Bi-LSTM and Transformer, leveraging multi-scale temporal contexts. However, further clarification is needed on the impact of different temporal context scales.

In the "EXPERIMENTS" section, the analysis on the consistency loss demonstrates its importance, as the AUROC metric drops by 1.80% without it. This supports the need for aligning different event-relevant prediction scores. However, more detailed reasoning on why this alignment improves performance would be beneficial.

In the "CONCLUSION" section, the summary effectively highlights the contributions and results of the study. However, it would be beneficial to include specific numerical results to quantify the improvement and reinforce the claims of state-of-the-art performance. Additionally, discussing potential future work or limitations could provide a more comprehensive conclusion.

**Suitability:**

3

---

### Official Review · Reviewer_V8H4 · 2024-05-24

**Rating:** 5
**Confidence:** 3

**Summary:**

The paper addresses the open-domain AVE (Audio-Visual Event) problem, which involves training a model using a labeled training set and then recognizing events in video data that have a different distribution from the training set. This problem avoids excessive reliance on the training data, offering broader practical applicability.

**Strengths:**

The paper achieves the final discrimination of video segments by introducing both event prediction and clustering simultaneously. By aggregating the results of the discriminative outcomes, the model avoids relying solely on the prediction module and excessively on true labels, which leads to poor model generalization.

**Limitations:**

In the classification module, the authors incorporate a multi-scale event perception module to maintain event continuity by jointly leveraging contextual features. However, applying multi-scale processing to all segments to be recognized reduces model efficiency. It might be worth considering increasing multi-scale computations only for uncertain segments based on the prediction module's outcomes.
The authors obtain new unknown class discrimination results by integrating features, which may lead to the erroneous recognition of complex backgrounds as events. It could be beneficial to perform discrimination across different modalities before fusion to gain complementary multi-modal information.

**Suitability:**

2

---

### Official Review · Reviewer_hNKo · 2024-06-02

**Rating:** 3
**Confidence:** 4

**Summary:**

Existing works on audio-visual event localization task are operated under a closed-set assumption and struggle to recognize unknown events in open-world scenarios. To this end, this paper studies the Open Set Audio-Visual Event localization (OSAVE) task and proposes the OpenAVE framework which is based on evidential deep learning. Experimental results demonstrate the effectiveness of the proposed task.

**Strengths:**

- The studied open set audio-visual event localization task is novel and valuable.
- In addition to the new task, this paper also contributes a strong baseline method that achieves satisfactory open-set event localization.

**Limitations:**

- Some contents in the paper should be more rigorous. For example, the videos in the AVE dataset are 10 seconds, which may not be 'untrimmed' as the authors claimed in Line 48. In Lines 102-104, the authors wrote that 'given an untrimmed testing video containing a unknown event (e.g., Baby Cry), traditional AVE methods fails to identify this unknown audio-visual event segment and tends to assign its label to a known class (e.g., Male Speech).' It would be better if there are some detailed experimental evidences. For example, the authors may select two or three traditional models and report their FAR@95 under the claimed scenarios.
- It would be better to include all the other evaluation metrics in Table 4. Besides, some references related to the datasets for feature extraction, baselines for comparison, and evaluation metrics should be cited appropriately.
- In Table 3, the authors compare the proposed MSCPM module with the Bi-LSTM and Transformer. In my opinion, the proposed method can also be adapted to be plug-and-play by replacing the MSCPM using existing AVE backbones. If so, it would be better to supplement more comparison results to enhance this part.
- The evidential deep learning used in the proposed method seems to be widely adopted in those temporal action localization [2, 10, 11, 18] methods. The key differences or insights should be emphasized more to demonstrate the novelty of the proposed method better.
- This paper currently contains many grammatical errors. I only list a few of them here: Line 421- 'these model', Line 555 - 'add a additional',  Line 6 - 'TxC+1', last sentence in the caption of Fig. 2.
- Since this paper involves three dataset splits, the source codes are essential for readers.

**Suitability:**

3

---

### Meta-Review · Area_Chair_G1WY · 2024-06-30

**Recommendation:** Accept (Oral)
**Confidence:** 4

**Metareview:**

All the reviewers are satisfied with the response. The final paper should incorporate the author's valid responses. I am delighted to recommend the acceptance of this paper.

---

### Meta-Review · Senior_Area_Chairs · 2024-07-10

**Recommendation:** Accept (Oral)
**Confidence:** 5

**Metareview:**

This paper received mixed ratings initially. After rebuttal, all the reviewers tend to accept the paper. SAC and AC agree with reviewers and recommend acceptance.